# Melatonin as Master Regulator in Plant Growth, Development and Stress Alleviator for Sustainable Agricultural Production: Current Status and Future Perspectives

**Khadija Nawaz** [1,†]**, Rimsha Chaudhary** [1,†]**, Ayesha Sarwar** [1,†]**, Bushra Ahmad** [2]**, Asma Gul** [2]**, Christophe Hano** [3]**, Bilal Haider Abbasi** [4] **and Sumaira Anjum** [1,*]

[1] Department of Biotechnology, Kinnaird College for Women, Lahore 54000, Pakistan; khadijanawaz95@gmail.com (K.N.); rimshachaudhary.09@gmail.com (R.C.); ayeshasarwar7890@gmail.com (A.S.)

[2] Shaheed Benazir Bhutto Women University, Peshawar-25000, Pakistan; bushraahmad@sbbwu.edu.pk (B.A.); asmagul@sbbwu.edu.pk (A.G.)

[3] Laboratoire de Biologie des Ligneux et des Grandes Cultures (LBLGC), INRAE USC1328, Université d'Orléans, 28000 Chartres, France; hano@univ-orleans.fr

[4] Department of Biotechnology, Quaid-i-Azam University, Islamabad 54000, Pakistan; bhabbasi@qau.edu.pk

\* Correspondence: sumaira.anjum@kinnaird.edu.pk; Tel.: +92-300-6957038

† These authors contribute equally.

**Abstract:** Melatonin, a multifunctional signaling molecule, is ubiquitously distributed in different parts of a plant and responsible for stimulating several physiochemical responses against adverse environmental conditions in various plant systems. Melatonin acts as an indoleamine neurotransmitter and is primarily considered as an antioxidant agent that can control reactive oxygen and nitrogen species in plants. Melatonin, being a signaling agent, induces several specific physiological responses in plants that might serve to enhance photosynthesis, growth, carbon fixation, rooting, seed germination and defense against several biotic and abiotic stressors. It also works as an important modulator of gene expression related to plant hormones such as in the metabolism of indole-3-acetic acid, cytokinin, ethylene, gibberellin and auxin carrier proteins. Additionally, the regulation of stress-specific genes and the activation of pathogenesis-related protein and antioxidant enzyme genes under stress conditions make it a more versatile molecule. Because of the diversity of action of melatonin, its role in plant growth, development, behavior and regulation of gene expression it is a plant's master regulator. This review outlines the main functions of melatonin in the physiology, growth, development and regulation of higher plants. Its role as anti-stressor agent against various abiotic stressors, such as drought, salinity, temperatures, UV radiation and toxic chemicals, is also analyzed critically. Additionally, we have also identified many new aspects where melatonin may have possible roles in plants, for example, its function in improving the storage life and quality of fruits and vegetables, which can be useful in enhancing the environmentally friendly crop production and ensuring food safety.

**Keywords:** melatonin; defense mechanism; gene regulation; growth; development; abiotic anti-stressor

## 1. Introduction

Melatonin (N-acetyl-5-methoxytryptamine), an indoleamine, was first isolated in 1958 from the bovine pineal gland [1] and named according to its role in reversing the darkening effect of the melanocyte-stimulating hormone [2]. It is synthesized by the catalysis of N-acetyl-5 hydroxy tryptamine-methyl transferase tryptophan decarboxylase, tryptamine-5 hydroxylase and 5-hydroxy tryptamine-N-acetyltransferase, which are then catabolized to 2-hydroxymelatonin. It is generally produced in the chloroplast and

mitochondria of roots and leaves, and is further transferred to the meristem, flowers and fruits in plants [3], while in vertebrates melatonin is mainly produce in the pineal gland and is secreted rhythmically into the blood stream after its production [4]. In animals it is produced at night as its concentration in the blood gradually is reduced during daytime [5]. For this reason the day-to-day profile of melatonin levels can signal the time, in order to notify cells about the daytime as well as the yearly season. Exogenous melatonin implication to cells can mimic the nighttime, and thus it affects diurnal rhythms and photoperiodic responses [6]. In animals, melatonin is predominantly involved in the regulation of processes like photoperiodism and circadian (24-h) rhythm and is also recognized as an extraordinary signaling molecule in stimulating immunomodulatory and cyto-protective properties in both animals and humans [6].

Early studies dealing with melatonin isolation described its role in detoxification of free radicals, generated by either metabolism or photosynthesis in both animals and plants [7]. Later experimentations and research revealed its pleiotropic nature, which can also act as an antioxidant with a special role in the control of reactive oxygen and nitrogen species (ROS, RNS) and in inflammation suppression in plants [8,9]. Melatonin has a more efficient capacity to scavenge ROS and RNS than glutathione and vitamin E, as well as to regulate certain antioxidant enzymes such as glutathione reductase (GSSG-R), catalase (CAT), peroxidase (POX) and superoxide dismutase (SOD). It increases the efficiency of the electron transport chain in mitochondria and thus reduces electron leakage [10,11].

First isolation and elucidation of melatonin in a photosynthetic organism such as *Gonyaulax polyedra* have stimulated the exploration of melatonin in other autotrophic species and in higher plants [12,13]. Melatonin is considered as a master plant regulator as it stimulates plant growth and development by acting as signaling molecule associated with defense mechanisms against different biotic and abiotic stresses like cold, drought, salinity and nutritional deficiencies [14,15]. It is also involved in multiple physiological actions such as photosynthesis, seed germination that occurs in plants [16]. Biotic and abiotic stresses in plants cause reductions in yield, growth, senescence and even sometimes death, but in return they produce different types of responses to their protection against stresses. Melatonin production is one of the important features in stress responses in plants, which has similar effect as of indole-3-acetic acid (IAA) to protect them [17,18].

Currently, research on plant melatonin is in an exponential growth phase and its functions in numerous wild and transgenic plants have been uncovered. The number of publications related to possible physiological and genetic effects of melatonin on plants has rapidly increased within the last decade. In this review we confine ourselves to a discussion of most relevant and recent studies on the prospective effects of exogenous and endogenous applications of melatonin in plants, such as its ability as a first-line defense against oxidative (biotic and abiotic) stress responses, and its role as plant growth promoter, in crop improvement, chlorophyll preservation, increasing photosynthetic activity and its action as a gene expression modulator. In addition, the potential role of melatonin as plant growth regulator, the underlying enzymatic and genetic mechanisms and limitations in effective implementation of melatonin in agriculture practices are commented on. Moreover, we also speculate on new potential aspects where melatonin may have possible functions in plants.

## 2. Production of Melatonin in Plants

Production of melatonin in plants is different from that in animals. Its synthesis is regulated by many factors in plants; one of the major factors is light. In plants the major sites for melatonin biosynthesis are mitochondria and chloroplasts. Different groups of enzymes are present in these organelles to synthesize melatonin via synthetic procedures. The production of melatonin is a two-way method, so if the melatonin production is blocked in mitochondria it will start in chloroplasts. In plants it is produced by a specific enzyme M3H [19].

The production cycle of melatonin starts with the synthesis of tryptophan from tryptophan decarboxylase (TDC). TDC converts into tryptamine, which then converts into tryptamine-5-hydroxylase (T5H). T5H catalyzes tryptamine to serotonin. In the end serotonin is converted into melatonin, which is catalyzed from tryptophan decarboxylase (TDC) and converts into tryptamine. Some plants have different pathways, such as *Hypericum perforatum*, in which biosynthesis starts from tryptophan that is catalyzed into 5-hydroxytryptophan from tryptophan-5-hydroxylase. After that, TDC/AADC (aromatic-L-amino-acid decarboxylase) converts 5-hydroxytrytophan to serotonin. Within two days, serotonin converts to N-acetyl-serotonin from serotonin N-acetyltransferase (SNAT)/arylalkylamine N-acetyltransferase (AANAT), and then N-acetyl-serotonin methyltransferase (ASMT)/hydroxyindole-O-methyltransferase (HIOMT) catalyzes N-acetyl-serotonin into melatonin. Other than this, tryptamine can also be formed from SNAT into N-acetyl-tryptamine, but that is unable to further convert into N-acetyl-serotonin by T5H. Still no pathway has been identified yet to convert N-acetyl-tryptamine into N-acetyl-serotonin. Another way is through which serotonin can be converted into 5-methoxy-tryptamine via HIOMT and, in conclusion, transform 5-methoxy-tryptamine into melatonin by SNAT [8]. A schematic presentation of the melatonin production cycle is illustrated in Figure 1.

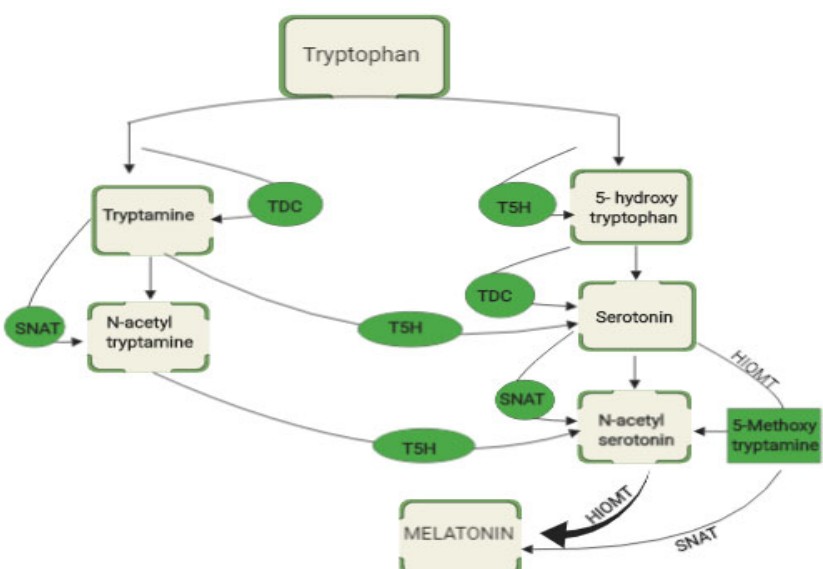

**Figure 1.** Biosynthesis of melatonin. Synthesis of melatonin starts with the production of tryptophan from tryptophan decarboxylase (TDC). TDC converts into tryptamine, which then converts into tryptamine-5-hydroxylase (T5H). T5H catalyzes tryptamine to serotonin. Lastly, serotonin is converted into melatonin, which is catalyzed from tryptophan decarboxylase (TDC) and converts into N-acetyl tryptamine.

Melatonin plays an important role in modulating the gene expression of plant hormones such as auxin carrier proteins and plays a role in the metabolism of auxins (IAA (indole-3-acetic acid), in particular), cytokinin, abscisic acid (ABA), gibberellins (GAs) and ethylene. It has been noticed that melatonin behaves like auxins such as IAA to promote the growth of lateral and adventitious roots. Melatonin shelters photosynthesis and postpones senescence [20].

### 3. Melatonin: Provoking Defense Mechanisms against Various Stresses in Plants

In addition to the excellent role of melatonin as an antioxidant molecule, melatonin is capable of highly efficient scavenging of free radicals against hazardous and deleterious reactive species like nitric oxide, hydrogen peroxide, singlet oxygen, hypochlorous acid

and many more, as shown in Figure 2. Further applications of melatonin are observed in providing a protective role and defense mechanisms against various abiotic stresses of cold, heat, drought, heavy metal, chemicals and ultraviolet radiation. Melatonin has the ability to provoke higher nitrogen and chlorophyll content, higher ability of photosynthesis, higher content of soluble proteins and rubisco. It has the ability to increase the upregulation of protein responsible for photosynthesis during leaf senescence. Melatonin can uptake the regulation of three chloroplast ATP synthases enzymes. It has a role in enhancing the photosynthetic ability of plants and can improve the efficiency of photosystem I and photosystem II during light and dark cycles. Further distinguished roles of melatonin under various stress conditions are described below with case studies [21].

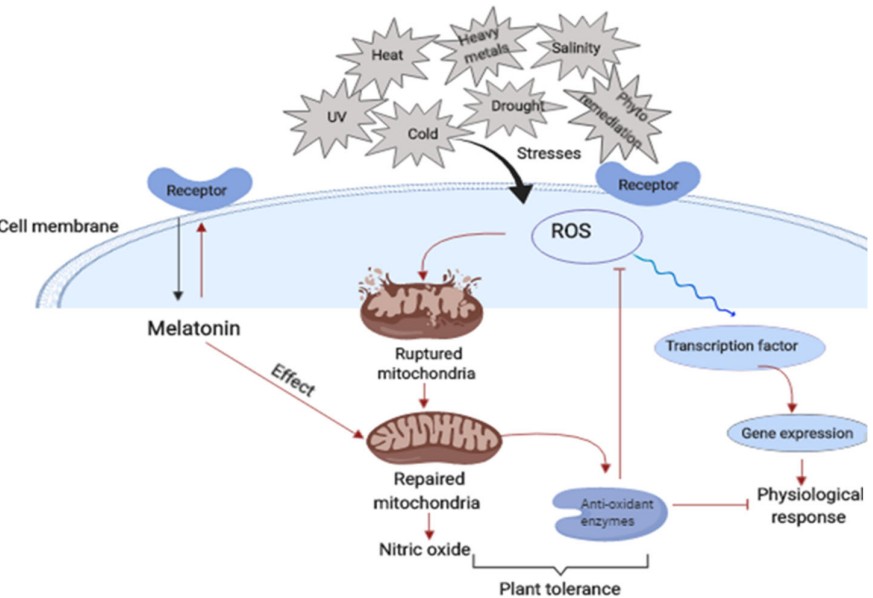

**Figure 2.** Melatonin as a pleiotropic molecule with various mechanisms of action in plants.

### 3.1. Salinity Stress

Due to the environmental stress of salinity, crop production and growth are limited, which results in great losses to the economy [22]. Salinity causes water deficiency due to the osmotic stress and disturbance in the regulation of biochemical cycles in plant cells like lipids and energy metabolism, protein synthesis and photosynthesis [23]. In order to cope with various type of stressors plants use a lot of different strategies including compartmentalization of ions, selective ions exclusion, altering photosynthetic pathways, compatible solutes production, antioxidant enzymes induction, alteration in membrane structure, upregulating gene expression and stimulating phytohormones [24]. Recently, in various plant growth regulators, auxin-independent melatonin effects are being observed. By the incorporation of 0.1 μM melatonin the plant growth inhibition caused by salinity stress was suddenly diminished, which proved significant in maintaining plant photosynthetic ability [25]. Furthermore, melatonin induction reduces oxidative stress, which is caused by production of ROS due to $H_2O_2$ scavenging, and shows great enhancement in activities of antioxidant enzymes like peroxidase, catalase and ascorbate peroxidase [23]. Salinity also exerts its obstructive effects on the plant, irrespective of growth retardation; it rapidly affects the plant from seed germination to the process of senescence throughout the life cycle of the plant. Salinity stress has severe negative impacts on plant growth and seed germination [24].

In cucumber, *Cucumis sativus* L., seed treatment with 1 μM melatonin significantly increased growth and germination rate from less than 150 mM NaCl stress; this

enhancement proved to be efficient in a 5-fold increasing of the activities of antioxidant enzymes like catalase, superoxide dismutase. Another role of melatonin is in the catabolism and biological synthesis of gibberellic acid and abscisic acid. It was shown that melatonin helped in downregulation of ABA biosynthetic genes and upregulation of ABA catabolism genes, which resulted in significantly decreased ABA. Meanwhile, it helps in upregulating biosynthetic genes of GAs in early germination stages, which resulted in better plant growth and seed germination during initial cycles [26]. In another example, induction of melatonin increased salt and drought stress tolerance in soybean and caused upregulation of genes that were hindered by salt stress [27]. In a recent experimental study, abiotic stress tolerance of melatonin was observed in Bermuda grass, and the study showed that 3933 transcripts (out of which 2631 showed upregulation and 1572 showed downregulation) showed significantly different expression than non-treated plants [28]. Melatonin was proved to be efficient in enhancing metabolic activity as it overexpressed genes involved in hormones, nitrogen and secondary metabolism, redox, transport and transformation of tricarboxylic acid. Furthermore, melatonin's role in combating salinity stress has been observed in various plants including *Cucumis sativus, Lupinus albus, Avena sativa, Vicia faba* L., *Brassica napus* L. and *Ipomoea batatas* L., which resulted in increasing overall plant growth and other morphological changes. Thus this proved the significance of melatonin as melatonin is able to remediate negative impacts due to salinity stress and enhances plant growth and seed germination [29].

### 3.2. Cold Stress

Agricultural crops are being damaged due to the cold stress as low temperature alters the molecular biology, physiology and biochemistry of plants. Scientists are focusing on development of crop cultivars that are cold-tolerant and are easily commercially available. Recent application of melatonin resulted in reducing impacts of cold stress in various plants.

When *Arabidopsis thaliana* plants were treated with 10–30 μM of melatonin, it resulted in increased length of root, fresh weight and height of plant as compared to untreated plants [30]. Cold stress to the wheat plant resulted in reduced area and water content of the leaf, photosynthetic pigment, and induction of ROS resulted in membrane lipid peroxidation. So the induction of 1 mM melatonin to the wheat seedlings for 12 h resulted in enhanced antioxidant activity of enzymes, glutathione reductase, superoxide dismutase and ascorbate peroxidase that caused increased plant growth by overcoming oxidative damage [31]. Recently, exogenous melatonin induction led to enhanced cold, drought and salt resistance in Bermuda grass. In this experiment, melatonin proved to be efficient because of not only antioxidant enzyme activation but also secondary metabolites were increased in high concentrations, like sugar, amino acid, alcohol and organic acids [28]. Furthermore, 100 μM of melatonin concentration when applied to *Camellia sinensis* L. resulted in mitigating cold-induced reduction in photosynthetic ability due to a decrease in oxidative stress. The enhanced antioxidant and redox homeostasis potential of melatonin thus helps in alleviating morphological changes caused due to cold stress [32]. Besides exogenous application of melatonin, an increase in endogenous levels of melatonin by using various transgenic approaches has gained popularity in combating various abiotic stresses in plants. Various enzymes (TDC, ASMT, AANAT, and HIOMT) responsible for regulating the melatonin biosynthesis pathway have been successfully overexpressed in various crops and showed good results by boosting the level of endogenous melatonin [33–35]. In line with this, the tomato lines overexpressing the ASMT gene improved the level of endogenous melatonin. Additionally, it increased the cellular protection through the generation of heat shock protein and activation of autophagy to refold denatured proteins that triggered heat resistance [36].

Altogether, these exogenous applications and endogenous boosting of melatonin levels in various plants like tomato, pea seedlings and rice have been reported, which has greatly influenced the plant's growth, morphological changes, photosynthetic carbon

fixation, photosystem II ability and overall activity of antioxidant enzymes. All these reports exhibited the antioxidant potential of melatonin in combating cold stress-induced changes.

### 3.3. Heat Stress

Due to the highly extreme temperatures enzyme activity and membrane fluidity of plants are severely affected, which results in altered development and growth of plant and thus loss of yield [34]. Under stress conditions in plants, genes that are responsible for inducing melatonin are activated, hence increasing the melatonin levels in plants. Melatonin has a role in enhancing the activities of metabolic enzymes of nitrogen, thus increasing nitrogen content and limiting ammonia levels at elevated temperature under heat stress.

In an example, the production of melatonin was enhanced in rice during heat stress conditions, proving melatonin's role in tolerance against heat stress [33]. High temperature in green micro-algae *Ulva* sp. resulted in enhanced levels of melatonin, thus confirming its functions in providing tolerance against heat stress [37]. The induction of melatonin has a role in reverting the inhibitive impacts of low and high temperatures upon thermo-sensitive and photosensitive *Phacelia tanacetifolia* Benth seeds [38]. In another experiment, melatonin application resulted in 60% enhanced germination level of *A. thaliana* seeds as compared to the control, which was the result of the antioxidant ability of melatonin [39]. After induction of melatonin in stress genes in Bermuda grass, a 16-fold overexpression in zinc finger, heat shock transcription factors, CBF/DREB genes and target genes was observed as compared to the control plants. Melatonin treatment in tomato plants helped in enhancing not only cell membrane stability but metabolic gene expression, plant growth and in mitigating damage to the antioxidant defense system caused by heat stress [40]. Further, melatonin induction in different plants including *Triticum aestivum* L. and Cucumber greatly enhanced antioxidant enzyme activity, transcription of stress-responsive genes and stabilized photosynthetic machinery, which resulted in alleviating structural changes in the plants caused by heat stress [41]. All these studies thus proved to be efficient in explaining defense mechanisms of melatonin against heat stress by mitigating damages caused to various plants.

### 3.4. Drought and Ultraviolet Radiation Stress

The prolonged drought conditions severely retard the growth, development and physiology of plants. The effect of ultraviolet radiation on the plants results in abnormal growth and even leads to the death of the plant. Therefore, in order to cope with these negative and deleterious effects of stress to the plant, induction of melatonin to the plant played a significant role in its protection [42]. Melatonin has proved to be protective against ultraviolet and drought stress. Also, during drought stress, melatonin has the ability to increase the functioning of stomata. When induction of melatonin is through the roots, it results in various optimized parameters including electrolyte leakage, chlorophyll and water content, increased photosynthetic efficiency and conductance of stomata openings [43]. Also, drought-stressed plants treated with melatonin showed half the amounts of ABA than those that were not treated with melatonin. This reduced $H_2O_2$ and ABA concentration helps in increasing the performance of stomata, thus coping with these stresses [44].

In case of drought stress tolerance, melatonin induction in *Moringa oleifera* L. enhanced growth parameters and quantitative and qualitative yield due to improved phenolic content of plant [16]. Micro-Tom transgenic tomato plants that overexpress *AANAT* and *HIOMT* (homologous ovine) genes exert great apical dominance loss and increased tolerance against drought [45]. Many other plants, including *Medicago sativa*, maize, wheat seedlings and apple being treated with melatonin, greatly increased their antioxidant defense system via enhanced activities of enzymes like catalase, peroxidase and morphological modifications, thus improving overall plant growth [46]. Melatonin

levels are comparatively higher in ozone-resistant plant species than in ozone-damaged species [47]. Likewise, melatonin levels are higher in high UV-exposed Alpine and Mediterranean species of plants as compared to the levels of those that inhabit less UV-exposed areas [48]. When roots of *G. uralensis* were exposed to UV-B radiation, increased concentrations of melatonin were observed. The melatonin role against UV-B was also studied by another scientist, who proposed that damage to DNA was highly diminished in *Nicotiana sylvestris* plant species when exposed to UV-B radiation as this transgenic specie expresses a synthesis gene of melatonin [49]. Overall these reports clearly focus on the defensive role of melatonin against both drought and ultraviolet stress in various plants.

### 3.5. Heavy Metal and Chemical Stress

Plants undergo various growth inhibition and morphological changes due to the presence of heavy metals and chemicals in their roots and shoots. Melatonin has a protective role against heavy metal stress in plants. It has the ability to inhibit the transport of heavy metals from roots to shoots of plants, thus reducing heavy metal accumulation. Melatonin acts as an antioxidant and signaling molecule, thus enhancing plant tolerance against abiotic stress. Melatonin has the ability to act against hazardous contaminants as a free-radical scavenger [29]. The potential of melatonin to mobilize toxic metals through phytochelatins, their transport, and sequestration adds to the general bio-stimulatory effect of melatonin on plants, resulting in a high degree of plant tolerance against heavy metals. Furthermore, the improvement in the absorption and metabolism of elements such as nitrogen, phosphorous, and sulfur helps the process of phytoremediation [50]. Although data are still limited, it seems that the presence of several stressors (e.g., metals and drought) synergistically induces the response of melatonin biosynthesis, which reinforces the overall response of the system. Beyond the use of transgenic plants that over-accumulate melatonin, the application of exogenous melatonin or the induction of its biosynthesis through environmental elicitors can be excellent strategies for phytoremediation purposes.

In an experimental study, 0.1 μM concentrations of melatonin in watermelon seedlings eventually controlled vanadium ion concentration among roots and shoots of the plant and also enhanced chlorophyll content, plant growth and photosynthetic assimilation [51]. Melatonin induction in maize plants reduced lead contamination caused by its accumulation under stress conditions [52]. This report was further more highly supported after induction of 5 μM melatonin in soil resulted in enhanced tolerance against copper ions in the pea (*Pisum sativum* L.) plant [53]. This proved that melatonin shows significantly high prevention effects against the death of pea plants in copper-contaminated soil. Similarly, in another study, pre-sowing melatonin treatment of red cabbage seed resulted in diminished toxicity of copper ions during seedling and germination growth [54]. The induction of 1 mM zinc sulphate resulted in a 6-fold increase in melatonin concentration in barley roots, proving the role of melatonin in providing protection against abiotic and chemical stressors [55]. The protective role of melatonin has been investigated against a single oxygen-generating herbicide (Butafenacil), and transgenic rice highly rich in melatonin showed less hydrogen peroxide and malondialdehyde. However, the same plant showed higher catalase and superoxide dismutase activities than those of control plants [56]. Upon these findings and some other experimental studies, scientists suggested that highly increased concentration of upregulated melatonin help in protecting plant species from extremely harsh conditions of various heavy metals and chemicals present in soil that are carcinogenic for the plant [57]. Thus, an integrated approach using biotechnology techniques should be considered in order to get the desired benefits of controlling plant growth and physiology against chemical and heavy metal stress [58].

### 3.6. Pathogen/Disease Resistance

One of the most important pathways present in all the eukaryotes are MAPK cascades that are extremely preserved in the inner transduction and the exterior signals that are important for the internal interactions. These are the signaling pathways present in almost all plants that play roles in abiotic stresses, which include UV light, cold, heat, osmotic pressure, plant injuries and the effect of heavy metals. They also work against the biotic stress from the effect of pathogens [59]. When affected by biotic and abiotic stresses, MAPK cascades are activated in plants. MAPKs have different intermediates that activate when required, e.g., MAPK1, MAPK2, MAPK3, MAPK4, MAPK5, MAPK6 and MAPK7. All these intermediates are activated by MAPKK [60]. Melatonin signals trigger the pathogen resistance that can be transcribed through the signaling pathway of MAPK.

Melatonin is also involved in the defense mechanism of plants against biotic factors (virus, bacteria and fungi) only through the upstream activation of MAPKK intermediates MAPK4/5/7/9 [61]. The defense mechanism responses are controlled by MAPK3/MAPK6 counter to pathogens. One of the famous microbe-associated molecular patterns of flagellin-derived flg22 and hairpin peptide is activated through signaling pathways of MAPK3, MAPK4, and MAPK6. On the other hand, MAPK4 behaves as a negative controller against the defense mechanisms [62]. Other studies showed that when flg22 was treated with MAPKs the pathogen was not dependent on calcium, kinase protein, reactive oxygen, hormones, salicylic acid, jasmonic acid, ethylene and other signals [63].

It was also noted that melatonin was used against pathogenic bacteria in treatment with *A. thaliana*. The pathogen commonly treated was Pst DC3000 through gene expression in comparison with other related genes, which included PR1, PR5, PDF 1.2, etc. [64]. Melatonin can prompt the SIPK transcription with Arabidopsis ortholog [65]. Furthermore, banana plants treated with melatonin induction resulted in upregulation of nine MaHSp90s transcripts after which they finally became resistant to *Fusarium oxysporum* cubens [66]. All these reports show melatonin's role in combating pathogenic stress to the various plants, thus improving their activity and growth.

### 3.7. Oxidative Stress

Melatonin works as an antioxidant and signaling molecule to improve the response to plant stresses, especially abiotic stresses. As an antioxidant, melatonin in combination with other plant hormones is used to control plant stresses, physiological, biochemical and cellular responses. It changes the permeability of cell membranes mediated by ion transporters, which control the opening and closing of stomata in plants. Thus, the antioxidant properties of melatonin make it work as an endogenous plant bio-stimulator for biotic and abiotic stresses [14].

Melatonin is of prime importance in improving the tolerance in plants for heavy metals such as copper, cadmium, and zinc. Another important heavy metal, vanadium, is present in the Earth's crust. Experimentation has been done on watermelon seeds to check their response to vanadium and the effect of melatonin on vanadium to improve the tolerance of watermelon seeds. Practical applications of melatonin on watermelon seedlings that were exposed to vanadium showed that the chlorophyll content increased up to 50 mg/L and improved photosynthesis as compared to the non-melatonin-treated plants. It suppresses the effect of pheide a oxygenase (PAO) and senescence-associated gene (SAG) transcript levels. Superoxide dismutase and catalase activities are also improved through melatonin treatment. It can be used to lessen the availability of vanadium to plants, which would increase plant growth and reduce plant stresses [51]. Excessive or deficient levels of iron in plants can stop the plant growth and inhibit the antioxidant activity and secondary metabolism in plants. For this melatonin can be used as an antioxidant to change the gene expression in plants, which can stimulate a decrease in iron level in plants. Thus, melatonin can be used to balance the iron content in plants,

which will maintain a balance in photosynthesis and chlorophyll production in many plants [67].

The reducing antioxidant properties of plants are associated with phenolics and flavonoids. Literature reviewed on melatonin showed its antioxidant processes in plants through ferric-reducing antioxidant power. When plants are treated with melatonin, the ferric-reducing antioxidant activity boosts SAR-stressed plants, which also comprises their antioxidant properties. Melatonin reduces the hydrogen peroxide and lipid peroxide levels in plants, which ultimately reduces the stresses and enhances growth and photosynthesis in plants. Acid rain is a limiting factor in crop production that can be overcome with the exogenous use of melatonin in plants [68].

Melatonin has the ability to improve the anti-oxidative properties and defense system of plants. *Carya cathayensis* is one such Chinese hickory plant that is commercially available and economical and famous for its nuts. The major drawback of growing this plant is that it takes a long time to reach its nut-growing phase. Its growth also gets retarded by environmental challenges such as drought and climate change. During its vegetative and reproductive phase grafting can be done to reduce the production time. So melatonin is used in the grafting of this plant under drought stress, which results in improved efficiency of plant photosynthesis and antioxidant defense system. The improved antioxidant defense mechanism enables the plant to enhance its defense against ROS and the total accumulation of soluble sugars while triggering proline. This shows that melatonin standardizes biological routes at catalytic and molecular levels [69]. As for all other biotic and abiotic stress responses, melatonin is primarily used to enhance defense mechanism against oxidative stress conditions, some of which are discussed below in Table 1.

**Table 1.** Role of melatonin in plants under various stress conditions.

| Concentration of Melatonin | Type of Stress | Plant Species | Effect on Plants | References |
|---|---|---|---|---|
| 1 μM | Salinity | *Cucumis sativus* | Increased growth and germination rate; 5-fold increase in activities of antioxidant enzymes like catalase, superoxide dismutase. | [26] |
| 0.1 μM | | *Lupinus albus* | Salinity stress diminished, which proved significant in maintaining plant's photosynthetic ability. | [25] |
| 50–100 μM | | Soybean | Increased salt and drought stress tolerance and enhanced seed germination. | [27] |
| 50 pg g$^{-1}$ | | Bermuda grass | Abiotic stress tolerance and upregulation and downregulation observed. | [28] |
| 1 mM | | Barley | Chlorophyll content increased 2-fold. | [70] |
| 50–150 μM | | Cucumber | Improved growth, photosynthetic ability and reduce oxidative damage by scavenging $H_2O_2$ or enhancing antioxidant enzyme activity. | [71] |
| 500 mM | | *Vicia faba L.* | Enhanced growth parameters, total phenolic and carbohydrate content, photosynthetic pigments, indole acetic acid and relative water content. | [72] |
| 1 μM | | *Brassica napus L.* | Not only enhanced root growth inhibition due to NaCl stress but also promoted seedling root growth. | [73] |
| 0.5 μM | | *Ipomoea batatas* L. | Stimulated triacylglycerol breakdown, fatty acid $\beta$- oxidation and energy turnover under salinity stress, thus improved $K^+/Na^+$ hemostasis and plasma membrane $H^+$–ATPase activity. | [74] |
| 0.1 μM | | *Malus hupehensis* Rehd. | Upregulated expression of ion-channel genes *MdNHX1* and *MdAKT1* in leaves, alleviated [23] growth inhibition and photosynthetic capacity and reduced oxidative damage by directly scavenging $H_2O_2$. | [23] |
| 100 μM | | *Avena sativa* | Improved morphological growth, photosynthetic ability and reduced antioxidant stress by upregulating ROS-scavenging enzymes. | [75] |
| 10–30 μM | Cold | *Arabidopsis thaliana* | Increased length of root, fresh weight and height of plant. | [30] |
| 100 μM | | Bermudagrass | Improved photosystem II performance and accumulation of metabolites. Also antioxidant activity increased. | [76] |
| 1 μM | | Wheat seedlings | Enhanced activity of enzymes glutathione reductase, superoxide dismutase and ascorbate peroxide, which eventually increased plant growth by overcoming oxidative damage. | [34] |
| 1 μM | | Tomato | Promoted photosynthetic carbon fixation, antioxidant potential, accumulation of metabolites, expressions of cold-responsive genes in cold-stressed tomato. Further, reduced cold-induced damage to plant. | [77] |
| 100 μM | | *Camellia sinensis L.* | Mitigated cold-induced reductions in photosynthetic capacity by decreasing oxidative stress due to enhanced antioxidant potential and redox homeostasis. | [32] |
| 5 μM | | Pea seed | Germination rate increased to 73.2% as compared to 53.7%. | [78] |
| 100 μM | | Rice | Alleviated growth inhibition, accumulation of reactive oxygen species and cell death due to cold stress. Physiological, biochemical and photosynthetic abilities increased, also antioxidant enzyme activity enhanced. | [79] |

| | | | | |
|---|---|---|---|---|
| 1mM | | Wheat seedlings | Increased leaf surface area, exerting strong mitigating effect on relative water content and high antioxidant activities. Further alleviated reduction in pigment content and improved physiological morphology. | [31] |
| 0.5 ng mL$^{-1}$ | | *Ulva* spp. | Tolerance against heat stress. | [37] |
| 0.3–6 µM | | *Phacelia tanacetifolia* | Reverted the adverse impacts of low and high temperature on thermo-sensitive and photosensitive plants. | [38] |
| 1000 µM | | *Arabidopsis thaliana* | 60% increase in germination rate, which reached up to 92.8% due to antioxidant ability of melatonin. | [39] |
| 100 µM | | Tomato seedlings | Alleviated damage to antioxidant defense system and enhanced plant growth. It further increased cellular membrane stability and metabolic gene expression. | [40] |
| 100 µM | Heat | *Triticum aestivum* L. | Enhanced antioxidant enzyme activities, thus modulating their defense mechanism, and stabilized photosynthetic machinery by increasing chlorophyll content. Also regulated transcription of stress-responsive genes. | [41] |
| 10 µM | | Tomato | Promoted cellular protein protection by decreasing insoluble and ubiquitinated protein level and enhancing heat shock proteins and autography to refold denatured proteins. | [36] |
| 50 pg g$^{-1}$ | | Bermuda grass | 16-fold overexpression in zinc finger, heat shock transcription factors, CBF/DREB genes and target genes as compared to control. | [28] |
| 20 µM | | *Arabidopsis thaliana* | 50% increase in survival rate as compared to control under heat stress. | [80] |
| 25–100 µM | | Cucumber seedlings | Net photosynthetic rate increased and intercellular $CO_2$ concentration decreased. | [81] |
| 1–100 µM | | Red cabbage | Diminished toxicity of copper ions during seedling and germination growth. | [54] |
| 17 ng g$^{-1}$ | | Green algae | Relieved cadmium-induced stress. | [37] |
| 0.1 µM | | Watermelon seedlings | Enhanced plant growth, chlorophyll content and photosynthetic assimilation. Further, it lowered vanadium concentration in roots and leaves by reducing their transport from root to shoot, thus making plant tolerant to stress. | [51] |
| 100 µM | | Wheat plant | Improved plant growth, pigments and regulated uptake of essential elements. cPTIO combined with melatonin treatment enhanced oxidative stress and reduced antioxidant enzymes. | [82] |
| 0.45–0.51 ng/g | Heavy metal | Tomato plant | HsfA1a induced Cd tolerance by activating *COMT1* gene transcription and melatonin induction upregulated expression of *HSPs*. | [83] |
| 0.1 mM | | Maize plant | Melatonin induced endogenous NO to mitigate Pb toxicity from maize plants, thus increasing their tolerance against stress. It further induced the antioxidant defense system of plants. | [52] |
| 100 µM | | Cucumber | Alleviated oxidative stress by promoting activity and transcripts of oxidative enzymes. It altered iron uptake and enhanced photosynthesis and biosynthesis of secondary metabolites. | [67] |
| 5 µM | | Pea plant | Enhanced tolerance to copper contamination and increased survival rate. | [57] |
| 50 µM | | Alfalfa and *Arabidopsis* seedlings | Decreased cadmium accumulation via modulating heavy metal transporter capacity of melatonin. Further, it re-established redox homeostasis via miR398. | [84] |
| 10 µM | Drought | *Medicago sativa* | Enhanced plant tolerant phenotype, chlorophyll fluorescence and stomatal conductance. Further, it made the plant tolerant to drought by regulating nitro-oxidative and osmoprotective homeostasis. | [85] |

| Concentration | Stress Type | Plant | Effect | Ref. |
|---|---|---|---|---|
| 1 mM | | Maize | A photoprotective and anti-oxidative agent which combined with melatonin resulted in maximum efficiency of photosystem II photochemistry, thus exerting defensive role against drought in maize. | [46] |
| 100 µM | | Apple | Enhanced nutrient uptake by increasing gene expression and also increased uptake, utilization and accumulation of $^{15}$N. Also positively affected growth and physiological parameters of *Malus*. | [86] |
| 500 µM | | Wheat seedlings | Enhanced drought tolerance due to increased antioxidant capacity, GSH and AsA formation by increasing related gene regulation. Further, stimulated epidermis cell enlargement and decreased membrane damage by maintaining grana lamella in chloroplast. | [87] |
| 100 mM | | *Moringa oleifera L.* | Increased growth parameter, yield quantity and quality by improving phenolic content, photosynthetic pigments and antioxidant enzyme system. | [16] |
| 100 µM | | Maize | Reduced reactive oxygen species burst and increased photosynthetic activity, thus allowing plant to tolerate drought stress while maintaining its growth. | [88] |
| 0.1 mM | | Apple leaves | Increased resistance of *Marssonina* apple blotch-fungal disease against *Diplocarpon Mali* by modulating activities of antioxidant and defense enzymes, pathogenesis and hydrogen peroxide levels. | [89] |
| 100 µM | | Banana | Melatonin upregulated nine *MaHSP90s* transcripts, which eventually made it resistant to *Fusarium oxysporum* cubens. | [66] |
| 1 mM | Pathogen/Disease resistant | Cucurbits and watermelon | Served as immune inducer and boosted plant immunity and suppressed pathogen growth. Further increased resistance against *Podosphaera xanthii* and *Phytophthora capsici.* Altered gene expression involved in pathogen-associated molecular pattern and effector-triggered immunity defenses, thus increased tolerance against stress. | [90] |
| 5 mM | | Potato | Inhibited mycelial growth, cell ultrastructure changes and reduced *Phytopthora infestans* stress tolerance. It further altered expressions of genes associated with stress tolerance and fungicide resistance, thus preventing potato late blight. | [91] |
| 10 µM | | *Arabidopsis thaliana* | Acted as defense signaling molecule in plants against pathogen. | [64] |

## 4. Melatonin: A Multifunctional Factor in Plants

A great literature review of melatonin has shown its importance in plants as a growth promoter with both in vitro and in vivo effects. In vitro effects refer to the growth of plants in response to exogenous treatments of melatonin (externally provided). Whereas, the in vivo effects refer to the growth and development of plants in response to melatonin produced endogenously. It was proved that melatonin is naturally produced endogenously in various plant species, including medicinal herbs, crops and fruit, although melatonin content was found to vary in different plants, as summarized by Fan and his co-workers in their study in 2018 [8].

The antioxidant nature of melatonin makes it useful as a bio-stimulant in agricultural processes by developing improved resistance against oxidant, biotic and abiotic stress response caused by scavenging ROS, and protection against bacterial pathogens [92]. All the defense mechanisms discussed in the above section have resulted in improved plant traits, quality and increased biomass. Melatonin helps in enhancing seed and root germination, flower formation and fruit development, with increased survival rates under harsh conditions. Higher levels of melatonin present in plants delay leaves' senescence, preventing degradation of chlorophyll, and also help in increased photosynthetic activity by maintaining homeostasis levels. All of these functions of melatonin in plant growth and development are discussed below in detail.

### 4.1. Melatonin as Growth Promoter in Plants

In the growth and development process of plants, various phytohormones are involved, especially the auxin. As a kind of indoleamine, melatonin shares the same initial compound, which is tryptophan with indole-3-acetic acid (IAA), so melatonin should play a role in the regulation of growth and development in plants [8]. Melatonin is regarded as a growth-promoting molecule due to its dual nature of promoting plant growth by increasing various growth parameters with both in vitro and in vivo responses and enhancing plants' yield by regulating ion homeostasis [16]. Melatonin is an independent plant growth regulator that can regulate its own biosynthesis and that of several other natural and synthetic plant growth regulators, such as auxin, abscisic acid, gibberellins, cytokinins, ethylene, polyamines, jasmonic acid and salicylic acid. It is an important modulator of gene expression related to plant growth regulators and can also mediate their activities. Many studies have suggested that the growth-promoting effects of melatonin on plants are comparable to other plant growth regulators [20,93]. Most of the studies performed have dealt with the auxin-like activity of melatonin, which is able to induce growth in shoots and roots and stimulate root generation in plants. Melatonin is also able to delay senescence, protecting photosynthetic systems and related subcellular structures and processes like cytokinins. Similarly, its role in fruit ripening and post-harvest processes as a gene regulator of ethylene-related factors was also described by many researchers [94–96]. In fact, multiple interrelations may exist between melatonin and many other plant growth regulators, therefore, to evaluate the exact potential of melatonin on plant growth, firstly we have to develop a complete understanding of all the changes in gene expression modulated by melatonin.

Melatonin has structural similarity to auxins, specifically IAA, and hence is involved in regulation of roots and shoots. So it plays the role of auxin to encourage vegetative growth in a variety of plant species [97]. Later on it was observed that IAA stimulated root organogenesis, and cytokinin-induced shoot organogenesis was repressed with various amounts of endogenous melatonin and repressors of the transport of serotonin and melatonin [98]. This proposition leads to the fact that melatonin acts as a probable regulator of plant growth and development [99]. Numerous studies have confirmed this proposition. In one such study the etiolated hypocotyls from *Lupinus albus* L. (lupin) were incubated in the presence of a great range of melatonin and IAA. Both of these composites were seen to be dispersed in lupin tissues in a similar concentration gradient and

produced an active elevation of growth at concentrations in the micro–molar range but the plant showed inhibitory growth effect at higher concentrations in integral and de-rooted lupins. The reversal effects on the application of melatonin and IAA were seen because the meristematic regions were removed [100].

### 4.1.1. In Vivo Effects of Melatonin on Growth Promotion

Plant growth promotion is seen by in vivo effects on the presence of a plant's internal levels of melatonin and other regulating hormones. Melatonin has different levels of expression, depending on plant species [8]. Many studies on in vivo effects on plant growth have been done for which researchers credited the detected hypocotyl growth to cell expansion in tissues where melatonin in combination with IAA played a functional role. Additionally, both of these indoles stimulated the formation of root primordial from pericycle cells with resulting modification in the pattern of spreading of adventitious or lateral roots, the time-course, the number and length of adventitious roots and the number of lateral roots [101].

According to one study melatonin formed the maximum number of hypocotyls with analogous values of IAA for root length in the entire range of tested concentrations. This same group of researchers stated analogous effects of melatonin, such as an active elevation of growth and a development of inhibitory effect at high concentrations, probably due to auxin-induced ethylene biosynthesis in monocotyledons, e.g., wheat, oat and canary grass. Predominantly the optimum level of growth elevation attained in coleoptiles with IAA was 100%, the optimum growth-enhancing effect of melatonin was 10% for oat, 20% for wheat, 32% for the canary grass, and 55% for barley coleoptiles, which the researchers viewed as a substantial auxinic effect [102]. In the same way, the root growth fluctuated between 56% to 86% for the canary grass and wheat, respectively, with the growth-enhancing effect in lupin tissues up to 63% in different bioassays [100,102]. Melatonin also promoted the extension of etiolated cotyledons of *Lupinus albus* L. to the same level as it was noticed for IAA earlier [103].

### 4.1.2. In Vitro Effect of Melatonin on Growth Promotion

Along with in vivo effect, in vitro melatonin mechanism has also been involved in plant growth and development. In in vitro mechanism, externally provided melatonin plays the same role as that of the plant's already existing phytohormones, such as IAA and auxin. Melatonin and indoleamine share the same initial compound, which is tryptophan along with IAA, and so they behave the same as auxin for plant development and growth [8,102]. As melatonin and IAA have structural similarity to indole derivatives, so melatonin can be processed to IAA or as an agonist of IAA in plant tissues due to the fact that melatonin can be transformed into 5-methoxyindolacetic acid (a compound that exhibits low auxin activity) in animals [104].

Externally coating the plant's seeds with different amounts of melatonin has been studied in the literature. In one such report soybean seeds were coated with a range of melatonin amounts, which significantly improved plant height, leaf growth, seed amounts and increased vegetative growth regeneration of adventitious and lateral roots in soybean [27]. In another experimentation, Afreen along with her co-workers explained the in vitro effect of melatonin that stimulated the vegetative growth and expansion of *Glycyrrhiza uralensis* Fischer in a dose-dependent manner. The outcomes achieved by this group of researchers exposed evidence that the amount of melatonin in their model plant elevated the plant growth as it grew older, with values that were four times higher for a 6-month-older plant than for the 3-month-old plant. They also observed that the concentration of melatonin in the plant providing the highest growth rate was in those that were grown under the red light effect, signifying that there was a relationship between the concentration of indole and the growth and development of the plant [105].

The same effects were also mentioned in another study, in the wild leaf mustard, i.e., *Brassica juncea* (L.) Czern. They observed that lesser concentrations of melatonin, i.e., 0.1

µM, showed a root growth-enhancing effect whereas higher concentrations, i.e., 100 µM, showed an inhibitory effect while a stimulatory effect was only measurable in younger seedlings. Endogenous free IAA levels also increased at low levels of melatonin, whereas at higher levels of melatonin there was no predominant enhancement in the levels of IAA, and as a result there was strong inhibition in the root elongation. This caused researchers to propose that the inhibitory effect of melatonin on root elongation and growth actually involves a mechanism that is irrelevant to the mechanism of IAA [106].

Like the abovementioned experiments, many others have been documented with different levels and conditions of melatonin. Interestingly, the concentrations of serotonin and melatonin are high in thidiazuron (TDZ)-treated leaf explants with auxin-transported action suppressors. The augmentation of the TDZ medium with lidocaine, a sodium channel blocker, led to elevated levels of serotonin and melatonin but not auxin and considerably reduced the rate of TDZ-induced regeneration in the explants [107]. Taking this into consideration, the levels of serotonin and melatonin as well as auxin elevated by exposure to TDZ stimulated the regeneration of explant, explaining the fact that melatonin acts either as a hormone autonomously or in combination with auxin and its own precursors and metabolites [54].

In plants the activity of melatonin through $Ca^{2+}$-calmodulin can be a probable mechanism of signaling, considering the $Ca^{2+}$-dependent action of auxin in numerous physiological responses. Melatonin has a noteworthy influence on the cytoskeleton in plants and, additionally, it has great affinity for the calmodulin [108]. It is also responsible for the protein kinase $Ca^{2+}$-dependent inhibition, via the interaction of calmodulin–kinase being related to reorganizations of the cytoskeleton, which denotes some of the most primitive effects that have been described for melatonin [97,109]. So, the literature on *Echinacea purpurea* (L.) explants explained that activation of calcium channels (that change polarity of cells) enhanced the levels of melatonin, along with an inhibition in TDZ-induced callus induction [107]. This explains the growth-enhancing in vitro effects of melatonin at low concentrations, while inhibitory results occur at higher concentrations [110].

### 4.2. Role of Melatonin in Crop Improvement

Melatonin plays a pivotal role in crop improvement, as it is used as a potential bio-stimulator for enhancement of crop yields, safety, and is ecofriendly. It is biodegradable and safer to be used in organic farming. The excessive intake of melatonin in plants can prevent peroxidation of plant products due to its antioxidant nature and thus increase the shelf life of plants and improve crop yields [111,112]. The main component of plants is chlorophyll, so its preservation and photosynthetic stimulation is of great importance in crop improvement. Horticultural crops are major sources of food and feed, which face a number of environmental challenges from fungal, bacterial or viral infections. Melatonin regulates anti-stress mechanisms to make these horticultural crops stress-free with high quality production rates [113].

It also improves the post-harvest preservation of several types of fruits and vegetables [114]. Exogenous melatonin can significantly reduce $H_2O_2$ content in roots, resulting in delayed post-harvest physiological deterioration (PPD) symptoms caused by damage during harvest and treatment, ultimately prolonging the preservation period of vegetables (onion, cabbage, cucumber, cauliflower, beans, carrot and pepper) and fruits (apple, banana, cherry, olive, grape, cranberry, kiwi, mulberry, pineapple, pomegranate, and strawberry) [115]. Gray mold is one of main diseases caused in apples by *Botrytis cinerea* during the post-harvest time, which significantly reduces the shelf life of the apple. So Cao and co-workers experimented on this and used 200 µM exogenous melatonin on an apple plant for 72 h incubation in order to inhibit gray mold. Melatonin significantly increased the shelf life of apples by inhibiting gray mold fungal infection [114]. Literature reviewed on post-harvest preservation activity of melatonin showed significant results in soaked bananas and other fruits. These findings provide a valuable scientific basis for

future research aiming at extending the shelf life of fruits and vegetables. Although exogenous melatonin can be used to increase the preservation period of post-harvest fruits and vegetables, it can also be of great interest to determine whether the shelf life of fruits and vegetables can be prolonged by increasing endogenous melatonin via a transgenic approach [116]. A lot of different case studies show the antioxidant and preservative effects of melatonin in plants under stress conditions, which directly contributes to an increase in crop yield [117].

### 4.3. Role of Melatonin in Chlorophyll Preservation

As described earlier, chlorophyll is a major component of photosynthesis in plants and it needs to be protected and preserved. There are two photosystems—photosystem I and photosystem II—that generate energy in the form of photons that are captured by chlorophyll and are used in the synthesis of carbohydrates. Environmental stress factors such as temperature and UV radiation generate ROS in plants and trigger damage in reaction center D1 protein binding and also damage chlorophyll [117]. Another major cause of chlorophyll damage is leaf senescence, which degrades chlorophyll and causes changes in plant hormone levels, damages molecular integrity, degenerates the cell wall and ultimately leads to plant death.

Melatonin serves as an antioxidant in plants and protects chlorophyll against degradation. It protects the plants against senescence (loss of chlorophyll), which occurs due to ROS/RNS, and regulates SAGs. Different experiments and studies have evolved its chlorophyll preservation activity in many plants. An experiment on barley leaves treated with a solution containing melatonin (1 mM) for 48 h showed a 2X higher chlorophyll content in melatonin-treated leaves than in control [118]. Similarly, cucumber plants that come under heat stress are preserved by the effect of melatonin [81]. Exogenous melatonin may preserve plants under heavy metal (Zn, Cd and Pb) stress. It preserves chlorophyll content in plants by the radical rummaging activity of indoleamine of melatonin [37]. Indoleamine of melatonin not only preserves chlorophyll under stress response but also increases the efficiency of photosystem II and elevates levels of ascorbic acid [119].

Another study showed the role of melatonin in transgenic rice plants. SNA (serotonin N-acetyltransferase) is a precursor of N-acetyl-serotonin, which is further converted to melatonin. Transgenic plants are treated and expressed with the human SNA gene under the ubiquitin promoter with the Agrobacterium-mediated method. The transgenic plants that were used had both serotonin and senescent detached leaves [120]. Melatonin causes a delay in the senescence condition of leaves under abiotic stresses in both monocots and dicots [50]. Further research has been done on the biological functions of melatonin in post-harvest senescence and in chlorophyll preservation that can be of great significance in agricultural science.

### 4.4. Role of Melatonin in Photosynthetic Activity

In plants melatonin is not only helpful in the preservation of chlorophyll but has also been found efficient in the increased photosynthetic activity of chlorophyll with maintained redox homeostasis [121]. It protects many horticultural crops, such as wheat, barley, sweet cherry and many others, by preventing damage in the photosynthetic apparatus through the elevated levels of melatonin [42]. Many experiments have been devised in order to demonstrate this statement. In one such experiment, cucumber seedlings of 25–100 $\mu$M were sprayed with a variable range of melatonin. It increased the photosynthetic rate in both seedlings under normal and heat stress conditions, respectively. However the $CO_2$ concentration was reduced due to the conversion by carbohydrates for increased photosynthesis [81].

Besides horticultural crops, many reports have been published recently on the effect of melatonin on increased photosynthetic activity of transgenic plants [122–124]. In one study, a group of researchers increased the production of endogenous melatonin in the chloroplasts of various transgenic lines of *A. thaliana*. As a result, their chloroplasts

exhibited an improved redox state with lower ROS levels under salt stress, compared to the wild-type. The high melatonin content and low levels of ROS resulted in all the transgenic plants being more tolerant to salt stress. This was indicated by the enhanced photosynthetic efficiency and higher biomass of the transgenic plants, compared to the wild-type under salt-stressed conditions [122]. Similarly, in another study, transgenic Bermuda grass treated with melatonin upregulated the expression of photosynthesis-related genes under salt stress, which resulted in the maximum increase in the photochemical efficiency of PSII and the total chlorophyll content by enhancing the biosynthesis of chlorophyll and slowing the rate of its decomposition. Therefore, melatonin plays a key role in protecting PSII and ameliorating the decrease of chlorophyll content under salt stress [28]. In line with the above findings, the influences of foliar-sprayed melatonin on maize seedling growth during drought stress were investigated in another study. Results showed that the melatonin-treated plants showed higher photosynthetic rates, stomatal conductances and transpiration rates as compared to untreated plants [125]. In addition to the abovementioned example many other experimentations and studies were reviewed for increased photosynthetic activity by melatonin.

### 4.5. Role of Melatonin in Increases in Biomass

Since melatonin is being used in crop improvement, another main factor is whether melatonin can increase the biomass of plants. Different modifications were made to the metabolic indoleamine enzyme of melatonin to develop transgenic plants [126]. Melatonin has the ability to increase the biomass of plants by improving efficiency in the growth of plants, developing defenses against stress response and improving the germination of seeds and seedlings. One such experiment was performed on red cabbage seedlings (*Brassica oleracea rubrum*) in which its seeds were incubated with melatonin at concentrations of 1, 10 and 100 μM with the hydropriming method. Both the control (non-incubated) and the pretreated seeds were allowed to germinate in darkness at 25 °C for three days. The young seedlings were then grown for five more days in light. Both the control and pretreated seedlings were tested with $CuSO_4$ water solutions at different concentrations. The seedlings were affected by Cu water, but the rate of germination in the non-treated was 53.5%; however, it was 73% in pretreated seeds. So the pre-treatment with melatonin of seeds increased the rate of germination in plants [54].

Corn plants treated with exogenous melatonin had irreversible generated genes for larger root systems. Melatonin increased by 20% production rates in corn plants [54]. *Arabidopsis thaliana* is one of the most important plants in many aspects. In *A. thaliana* the biomass was increased under the effect of serotonin (a precursor of melatonin) on the lateral root development system analyzed by using GC-MS (gas chromatography coupled to mass spectrometry). It increased around 10–160 μm in the root development system. The increase in concentrations of serotonin stimulated the primary root growth and root hair development. All these results show that exogenous concentrations of serotonin are able to convert into melatonin, and it can antagonize the higher levels of serotonin concentrations [127].

Mechanisms that define how melatonin promotes root development are still under research. Melatonin is highly present in different parts of the root with an obvious gradient. Its order of presence and quantification was done by liquid chromatography along with fluorescent detection. It was highly seen in the apical, central and then at the basal portion of the root system. With the help of liquid chromatography not only was the presence of melatonin detected, but also the changes in the melatonin level due to the influence of light and dark reactions were determined [128].

As a consequence, all of the above effects of melatonin, like chlorophyll preservation in leaves, root development, root regeneration, increase in biomass, delayed senescence in leaves, directly contribute to crop improvement and ultimately lead to increased food production. Most of the world is facing malnutrition and increased food demands with

low resources, so in this scenario melatonin contributions to crop improvement will play a great role. Food shortages of the commonly used crops of rice, wheat, barley and corn are a great problem for the world. Some of the other contributions of melatonin to crop improvement are by

- Increasing the rate of plant germination,
- Making plants resistant to environmental stresses,
- Gene manipulation of synthetic enzymes of melatonin, such as AANAT/ASMT.

Manipulation in melatonin genes of synthetic enzymes can be beneficial to the enhanced production of endogenous melatonin in plants. Moreover, the uses of agrochemicals such as BTH (benzothiadiazole) and chitosan, which activate the plant's own defense mechanism, are also greatly contributing in melatonin production [129]. Therefore, any method to meet the shortage of food with enhanced production will be a massive ramification. In this regard, melatonin modifies transgenic seed or seedlings with indoleamine and enhances crop production, while also improving the nutritional values of crops [126].

### 4.6. Other Functions of Melatonin in Plants

As explained above some of the other important functions of melatonin have also been interestingly reviewed and discussed. Melatonin in plants produces dark and light signals, which synchronize with the photoperiodic ecological that signals the daylight responses of flowering that only proceed in daylight with a short time tenure. *Chenopodium Renbur* L. is one such plant that has the ability of diurnal cadence that takes place at night [130]. Exogenous melatonin regulates the transition of flowering in many plant types such as *Chenopodium rubrum* L., *Pterygpra californica* Ruprecht, and *Arabidopsis thaliana* [131].

It plays an essential role in the maintenance of reproductive physiology and flower development of the family *Hypericum perforatum* L. (St John's wort). As during the uninucleate microsporogenesis stage, there are high concentrations of indole and melatonin contents, which lead to an increased regenerative potential of isolated anthers [132]. It plays a key role in the conservation of dormancy germs and in the preservation of different states in fruit tissues. Increased rates of melatonin in juicy fruits indicate its role in the ripening of fruits. Melatonin is essential in the regulation of growing stages that may lead to the conservation of dormancy. It has some specific actions on the chlorophyll degrading enzyme, named as chlorophyllase, pheophorbide that are oxygenase or red chlorophyll catabolite reductase and restricts the generation of free radicals, thus leading to a suspension in the senescence process [98].

Melatonin and other hormones in plants have hermetic biphasic expression of molecules and hormones, which are interrelated with the stress system of the plants. The adaptive significance of circadian rhythms displays high plasticity for managing the endogenous clock with the environment, which can be day or night duration. Cardiac rhythms and hermetic-type doses have dependent mechanisms. Other than that, plasma melatonin and cardiac rhythms present a U-shape pattern that gives a fully synchronized function of time. Another major function of melatonin relates to the emission of green lead volatiles with biosynthetic enzymes regulated by the cardiac clock [133].

Melatonin can be extracted from food or generated endogenously. It has antioxidant properties in both food and endogenous compounds and so can be taken in appropriate amounts from diet and other different living organisms. Excess production of melatonin can cause aging in plants as well as animals [110]. Daily intake of melatonin-rich foodstuffs such as coffee, corn, rice, tea, etc., can improve the health of consumers [126].

### 5. Conclusion and Future Perspectives

Transgenic organisms are the latest trends of scientific technology and hence are the most demanding in future needs. Transgenic plants have increased levels of melatonin in

genes induced from vertebrates. Transgenic crops will be produced in the future and the alterations in the synthesis of melatonin in these plants are essential for the induction of resistance against biotic and abiotic pressures, which results in the increase of crop production. In view of the significance and the satisfying value of melatonin for humans, the pharmaceutical industries should develop it for the preparation of pharmaceuticals for humans, animals and plants. There are some reports regarding the feasible role of melatonin in the control of insects and diseases. This information is the leading cause of usage of melatonin on a commercial scale. Melatonin causes the promotion of root growth but its role in the uptake of nutrients still requires investigation, since no reports have been published for the clarification of interactions between the melatonin and the uptake of nutrients and also about their transport. Likewise, very restricted information is available regarding the effects of foliar applications of melatonin on the consumption, growth, and development of plants. However, melatonin is universally present in plants, but it has not yet been discovered whether all the plant organs manufacture this indoleamine. The process of its transport across the whole plant must be discovered. Many scientists have discovered the auxin-like behavior of melatonin including its pleotropic functions too.

Melatonin being equally important in plants, animals and human beings can also pose several economic benefits to its users. As an antioxidant, the high levels of melatonin in plants are beneficial to its consumers, including animals and humans. Melatonin levels in popular beverages, including coffee, tea, red wine, and beer, and several major agriculture crops, including corn, rice, and wheat, are sufficiently high to raise blood melatonin levels after their consumption. Throughout the world, billions of people depend on these products as a major food source, so the potential health and economic beneficial effects of melatonin consumed in these products are obvious. Based on the data available and preliminary observations, it is hypothesized that melatonin application as a bio-stimulator can be a good, feasible and cost-effective method useful in agriculture. We also believe that this compound can increase the food quality (the aspect of functional food) and may improve the human health.

Numerous studies with melatonin have resulted in a set of data that indicates the excellent beneficial effects that this compound has on plants, especially in stress situations. However, the effective implementation of melatonin in agriculture practice is still limited due to several reasons, such as: (1) With respect to human health, it is classified as a non-hazardous substance in the oral, dermal, inhalation and irritation categories, and in regards to mutagenicity and carcinogenicity. However, melatonin is classified as a health hazard substance (code H-361) in terms of reproductive toxicity by EC (European Community) regulation, because it is suspected of damaging fertility or an unborn child. This classification reflects one of its multiple functions as an animal hormone, in which its participation in the modulation of sexual behavior and fertility in mammals has been demonstrated; (2) lack of information regarding the physiological and genetic effects of melatonin on plants; (3) since melatonin is an amphipathic molecule that crosses biological membranes and no adjuvant is needed, therefore, high concentrations of melatonin can pose toxic effects to humans and the environment; (4) little information is available on its effect on bacteria and fungi, especially those that are part of the soil microbiota (rhizosphere); (5) no data is present on its possible concentration-dependent toxicity; (6) it is expensive and, to date, no efficient strategies have been developed for extraction of melatonin from plants. All these limitations can be overcome by extensive research in the relevant field to find the answers to all queries related to effective implementation of melatonin in the agriculture sector.

In addition, more detailed investigations should be done regarding the feasible roles of melatonin in in vitro propagation of plants; processes of propagation through cuttings, vascular reunion and grafting; growth of flowers and their development; enhancement of male-to-female ratios in vegetables; betterment of fruit setting, parthenocarpy, fruit drops; breaking the tuber dormancy; quality of fruits; development of seeds and many other

processes, as they all need clarification. Root treatment of melatonin is essential for the improvement of the success ratio and growth of crops that need transplanting from nurseries. Keeping in mind all the beneficial effects of melatonin and their importance in biochemical, physiological, genetic and epigenetic processes in multiple organisms, it seems that melatonin is an important molecule for its influence on field crops, and has also proved to be essential in the enhancement of yields of crops and the nutraceutical values functional for addressing the food security issues across the whole world.

**Author Contributions:** Conceptualization, S.A., B.H.A. and C.H.; methodology, R.C., K.N., and A.S.; software, A.G. and B.A.; validation, B.H.A., S.A. and C.H.; formal analysis, S.A., R.C. and C.H.; investigation, R.C., K.N., .H., B.H.A. and S.A.; resources, S.A. and C.H.; writing—original draft preparation, R.C., K.N. and A.S.; writing—review and editing; S.A., C.H. and B.A.; visualization, R.C., K.N., A.G., C.H. and B.H.A.; supervision, S.A. and C.H.; project administration, S.A.; funding acquisition, C.H., S.A. and B.H.A.; All authors have read and agreed to the published version of the manuscript.

**Funding:** This research received no external funding.

**Informed Consent Statement:** Not applicable**.**

**Acknowledgments:** S.A. acknowledges the administration of Kinnaird College for Women, Lahore, for providing support.

**Conflicts of Interest:** The authors declare no conflict of interest. The funders had no role in the design of the study; in the collection, analyses, or interpretation of data; in the writing of the manuscript, or in the decision to publish the results.

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
