# Peer review of "Melatonin as Master Regulator in Plant Growth, Development and Stress Alleviator for Sustainable Agricultural Production: Current Status and Future Perspectives"

_sustainability, doi:10.3390/su13010294_

Round 1

Reviewer 1 Report

In my opinions:

this work is extensive and very interesting, describes the most important aspects of the effects of melatonin on plant growth and development.

A very interesting point of this work is 3.6. - the mechanism by which plants transmit signals to another neighbor plant of the same species when the pathogen invades is unknown (Fusarium for example).

I have one question to autor:

in the cultivation of flax (Linum L.), the plants warn each other. Can the authors in point 3.6 try to explain more whether melatonin is involved in signal transduction and leads to the synthesis of defense substances when the pathogen has not yet reached the plant but only neighbor plant around.
(?)

I dont have any other suggestion for this work.

Author Response

Reviewer-1

Comments: This work is extensive and very interesting, describes the most important aspects of the effects of melatonin on plant growth and development.

A very interesting point of this work is 3.6. - the mechanism by which plants transmit signals to another neighbor plant of the same species when the pathogen invades is unknown (Fusarium for example). I have one question to author:

In the cultivation of flax (Linum L.), the plants warn each other. Can the authors in point 3.6 try to explain more whether melatonin is involved in signal transduction and leads to the synthesis of defense substances when the pathogen has not yet reached the plant but only neighbor plant around(?)

I don’t have any other suggestion for this work.

AUTHORS: Thank you very much for your comments. Melatonin is an independent plant growth regulator but many publications showed that it can interfere and/or regulate the production and/or signaling of many key regulators such as calcium, reactive oxygen, or phytohormones involved in defense response such as abscisic acid, salicylic acid, jasmonic acid, ethylene as pointed in this paragraph 3.6.  One of our main hypotheses, to answer to the reviewer question about how “the plant (cell)s warn each other” involved the volatile stress phytohormone methyl jasmonate. Some of our unpublished results support the involvement of this phytohormone for that purpose, and a cross-talk between MEL and MeJA in defense response. Note that several publications presented in this review (not only in paragraph 3.6) also pointed this possibility.

Reviewer 2 Report

The review of “Melatonin as Master regulator in Plant Growth, Development and Stress Alleviator for Sustainable Agricultural Production: Current Status and Future Perspectives” for Sustainability MDPI.

The type of article is a review and the authors should not only summarize the current state of research on the functions and roles of melatonin in plants and the possible implementation of melatonin in agricultural practice, but also critically analyze and discuss the main contradictions giving various data on research problems.

In this regard, there are several comments that the authors must take into account before the manuscript is suitable to be published:

1) It should be noted that over the past 3 years, several review articles have been published on melatonin and its function in plants, including under the action of different abiotic and biotic factors (Some examples: Fan, J.; Xie, Y.; Zhang, Z.; Chen, L. Melatonin: A Multifunctional Factor in Plants. Int. J. Mol. Sci. 2018, 19, 1528; Tan, D. X., & Reiter, R. J. An evolutionary view of melatonin synthesis and metabolism related to its biological functions in plants. Journal of Experimental Botany, 2020; Moustafa-Farag, M.; Almoneafy, A.; Mahmoud, A.; Elkelish, A.; Arnao, M.B.; Li, L.; Ai, S. Melatonin and Its Protective Role against Biotic Stress Impacts on Plants. Biomolecules 2020, 10, 54; Debnath, B., Islam, W., Li, M., Sun, Y., Lu, X., Mitra, S., ... & Qiu, D.. Melatonin mediates enhancement of stress tolerance in plants. International journal of molecular sciences, 2019, 20(5), 1040). Therefore, the authors should clearly indicate in the introduction which new aspects will be discussed in their article and why it is at present time necessary/important to publish their review.

2) L193: Please explain what is meant by “endogenous application of melatonin”

3) L 215-217: Please check the phrase “caused by cold stress”. Probably it should be “heat stress”?

4) L 255: It has ability in inhibiting the transport of heavy metals from roots to shoots of plants”. The authors could give an idea about the eventual mechanism of such inhibition.

5) L 259: “extremely high concentrations of melatonin”. Please indicate the concentration of melatonin.

6) L281-284: When considering abiotic stresses above, nothing was said about MAP kinases, the possible role of melatonin in these cascades and this mechanism of increasing plant resistance to abiotic factors. Does melatonin involved in MAPK signaling pathways in plant growing under abiotic stress. If yes, please add the information to the appropriate subsections (3.1-3.5).

7) Subsection 3.7. Oxidative stress. Is this section really necessary? Above, when considering abiotic factors, there is a lot of information about a decrease in ROS concentration and changes in the activity of antioxidant enzymes through melatonin treatment. Additionally about the antioxidant properties of melatonin was discussed in the first paragraph of Section 3 and presented on the Figure 2.

8) L 353: Please clarify here what you mean about” in-vitro and in-vivo effects”. In reference [93] there are also no explanations about these terms. It is important to explain the use of these terms at the beginning of the Section because it is not clear further on their use in the Subsections 4.1.1 and 4.1.2. In both Subsections studies related to exogenous melatonin supplementation are presented. In both Subsections the possible mechanism of the effect of melatonin on growth processes through its connection with the hormone IAA is discussed. In the Subsection 4.1.2 “In vitro effect of melatonin on growth promotion” studies that were performed not only with plants cultivated in vitro, but also with plant  are presented (e.g. data on soybeans, Glycyrrhiza uralensis).

9) L 414: Check the reference [67]. In this paper investigation of soybean and not L. albus is presented.

10) L465: Please check the reference [113] and add some examples of “vegetables and fruits”.

11) L470: Please check the reference [113].

12) L 522: Please explain in more detail about transgenic plants and give some examples. Below in this Subsection, experiments with exogenous melatonin, and not with transgenic plants are discussed.

13) L 531: Please check the reference [119]. There is no experiment performed on B. rubrum  in this paper (119. B Arnao, M.; Hernández-Ruiz, J. Role of melatonin to enhance phytoremediation capacity. Applied Sciences 2019, 9, 5293).

14) L623: This is last sentence in conclusion, it is not good that the authors refer to another review of 2016 (Reference [29]), and do not make their own conclusion, based on the analysis and discussion of the data obtained since that time.

15) L624-631: text should be deleted

16) In general, the manuscript provides detailed data on the positive effect of melatonin on various parameters of plant growth and quality; however, there is practically no critical discussion. Below are some problems that could be additionally discussed by authors: What are the limitations that can restrict the effective implementation of melatonin in agricultural practice? How economically beneficial is its use? Is growth promoting effect of melatonin comparable to synthetic growth regulators? The answer to the last question is especially important in view of the aims and scopes of the journal Sustainability where the manuscript is submitted.

Author Response

Reviewer-2

Comments: The review of “Melatonin as Master regulator in Plant Growth, Development and Stress Alleviator for Sustainable Agricultural Production: Current Status and Future Perspectives” for Sustainability MDPI. The type of article is a review and the authors should not only summarize the current state of research on the functions and roles of melatonin in plants and the possible implementation of melatonin in agricultural practice, but also critically analyze and discuss the main contradictions giving various data on research problems. In this regard, there are several comments that the authors must take into account before the manuscript is suitable to be published.

AUTHORS: Thank you very much for your comments and suggestions that greatly help us to improve the quality of the present revised version of our MS. We do our best to answer to all your queries and hope this revised version will answer them. Revisions appear in track changings in the revised manuscript.

  1. It should be noted that over the past 3 years, several review articles have been published on melatonin and its function in plants, including under the action of different abiotic and biotic factors (Some examples: Fan, J.; Xie, Y.; Zhang, Z.; Chen, L. Melatonin: A Multifunctional Factor in Plants. Int. J. Mol. Sci. 2018, 19, 1528; Tan, D. X., & Reiter, R. J. An evolutionary view of melatonin synthesis and metabolism related to its biological functions in plants. Journal of Experimental Botany, 2020; Moustafa-Farag, M.; Almoneafy, A.; Mahmoud, A.; Elkelish, A.; Arnao, M.B.; Li, L.; Ai, S. Melatonin and Its Protective Role against Biotic Stress Impacts on Plants. Biomolecules 2020, 10, 54; Debnath, B., Islam, W., Li, M., Sun, Y., Lu, X., Mitra, S., ... & Qiu, D.. Melatonin mediates enhancement of stress tolerance in plants. International journal of molecular sciences, 2019, 20(5), 1040). Therefore, the authors should clearly indicate in the introduction which new aspects will be discussed in their article and why it is at present time necessary/important to publish their review.

AUTHORS: Sir our review article is very different from the above listed articles in many aspects. In this review article, we have critically analyzed the prospective effects of exogenous and endogenous applications of melatonin in plants along with the potential role of melatonin as plant growth regulator, the underlying enzymatic and genetic mechanisms and limitations in effective implementation of melatonin in agriculture practices.  Moreover, we also speculate on new potential aspects where melatonin may have possible functions in plants. On your suggestion, we have also added a paragraph stating the novelty of our article at Line 77-88.

  1. L193: Please explain what is meant by “endogenous application of melatonin”

AUTHORS: Sir, we have explained it in line 199-207.

  1. L 215-217: Please check the phrase “caused by cold stress”. Probably it should be “heat stress”?

AUTHORS: Sir we have corrected it in line-231, it was heat stress.

  1. L 255: It has ability in inhibiting the transport of heavy metals from roots to shoots of plants”. The authors could give an idea about the eventual mechanism of such inhibition.

AUTHORS: Thank you very much Sir for your valuable suggestion. We have floated an idea about eventual mechanism of inhibition on scientific background in line 272-281 in revised manuscript.

  1. L 259: “extremely high concentrations of melatonin”. Please indicate the concentration of melatonin.

AUTHORS: Sir we have added the concentration of melatonin in line 282.

  1. L281-284: When considering abiotic stresses above, nothing was said about MAP kinases, the possible role of melatonin in these cascades and this mechanism of increasing plant resistance to abiotic factors. Does melatonin involved in MAPK signaling pathways in plant growing under abiotic stress. If yes, please add the information to the appropriate subsections (3.1-3.5).

AUTHORS: Sir, melatonin is not involved in MAPK signaling pathways in plant growing under abiotic stress. Melatonin involved in MAPK signaling pathways in plant growing under only biotic stresses such as viruses, bacteria, fungi and other disease related pathogens therefore, nothing was discussed about MAPK in section 3.1-3.5, which are about abiotic stresses.  

  1. Subsection 3.7. Oxidative stress. Is this section really necessary? Above, when considering abiotic factors, there is a lot of information about a decrease in ROS concentration and changes in the activity of antioxidant enzymes through melatonin treatment. Additionally about the antioxidant properties of melatonin was discussed in the first paragraph of Section 3 and presented on the Figure 2.

AUTHORS: Thank you Sir for this remark. No doubt we have discussed the oxidative stress created in response to abiotic stress in plants in section 3, but the section 3.7 is different from the other ROS data in many aspects such as the antioxidant role of melatonin in production of valuable secondary metabolites (phenolics and flavonoids) and their effect on human health has been discussed. Therefore, we did not remove this section.

  1. L 353: Please clarify here what you mean about” in-vitro and in-vivo effects”. In reference [93] there are also no explanations about these terms. It is important to explain the use of these terms at the beginning of the Section because it is not clear further on their use in the Subsections 4.1.1 and 4.1.2. In both Subsections studies related to exogenous melatonin supplementation are presented. In both Subsections the possible mechanism of the effect of melatonin on growth processes through its connection with the hormone IAA is discussed. In the Subsection 4.1.2 “In vitro effect of melatonin on growth promotion” studies that were performed not only with plants cultivated in vitro, but also with plant are presented (e.g. data on soybeans, Glycyrrhiza uralensis).

AUTHORS: Sir we clear the main difference between in-vitro (exogenous) and in-vivo (endogenous) applications of melatonin in plant growth in line 378-383 in revised manuscript. Moreover, in the subsection 4.1.2 both wild and in vitro grown plants are discussed because “in-vitro” meaning here is the exogenous applications of melatonin not the “in-vitro production of plants in laboratory”.

  1. L 414: Check the reference [67]. In this paper investigation of soybean and not L. albus is presented.

AUTHORS: Sir we have checked the reference 67 now it is 71 in revised manuscript. We have corrected it in line 462.

  1. 10. L465: Please check the reference [113] and add some examples of “vegetables and fruits”.

AUTHORS: Sir we have added examples of fruits and vegetables in lines 513-515 in revised manuscript.

  1. 11. T L470: Please check the reference [113].

AUTHORS: Sir we have checked the 113 reference, it was mistakenly cited. We have corrected it, now it is 117 references in revised manuscript.

  1. L 522: Please explain in more detail about transgenic plants and give some examples. Below in this Subsection, experiments with exogenous melatonin and not with transgenic plants are discussed.

AUTHORS: Thank you for valuable suggestion. We have added data about transgenic plants in this subsection in line 568-580 in revised manuscript.

  1. L 531: Please check the reference [119]. There is no experiment performed on B. rubrum in this paper (119. B Arnao, M.; Hernández-Ruiz, J. Role of melatonin to enhance phytoremediation capacity. Applied Sciences 2019, 9, 5293).

AUTHORS: Thank you for pointing this mistake. We have corrected this reference and added the correct one (now reference 55 in revised manuscript). 

  1. L623: This is last sentence in conclusion, it is not good that the authors refer to another review of 2016 (Reference [29]), and do not make their own conclusion, based on the analysis and discussion of the data obtained since that time.

AUTHORS: Sir, all data present in conclusion and future perspective section is our own critical analysis and evaluation of already published data. The reference 29 was mistakenly cited there. We have removed that reference.

  1. L624-631: text should be deleted.

AUTHORS: Sir, we have deleted the text.

  1. In general, the manuscript provides detailed data on the positive effect of melatonin on various parameters of plant growth and quality; however, there is practically no critical discussion. Below are some problems that could be additionally discussed by authors: What are the limitations that can restrict the effective implementation of melatonin in agricultural practice? How economically beneficial is its use? Is growth promoting effect of melatonin comparable to synthetic growth regulators? The answer to the last question is especially important in view of the aims and scopes of the journal Sustainability where the manuscript is submitted.

AUTHORS: Sir, we have improved manuscript according to your suggestion. Your all questions are addressed in revised manuscript. Q.1 “What are the limitations that can restrict the effective implementation of melatonin in agricultural practice?” is addressed in conclusion section at line 693-705. Q.2 “How economically beneficial is its use? is also addressed in conclusion section at line 682-691.  Q.3 “Is growth promoting effect of melatonin comparable to synthetic growth regulators? This question is addressed in subsection 4.1 in line 401-414.

Round 2

Reviewer 2 Report

The authors improved the manuscript well. I am satisfied with most of the authors' responses to comments, but I have still one remark:

In my opinion, there is a contradiction between the two paragraphs in Lines 674-684 and in Lines 685-697 regarding the health benefits of melatonin for human and animals. In the first paragraph the authors write that “As an antioxidant, the high levels of melatonin in plants are beneficial to its consumers including animals and human” and “may improve the human health”. And in second paragraph, the authors write that “implementation of melatonin in agriculture practice is still limited due to …. (1) melatonin is classified as a health hazard substance (code H-361) in terms of reproductive toxicity by EC (European Community) regulation…”. I suggest to correct the phrase “(1) melatonin is classified as a health hazard substance (code H-361) in terms of reproductive toxicity by EC (European Community) regulation” by adding information about the non-hazardous effect of melatonin with respect to human health, in general, and mutagenicity and carcinogenicity, in particular.

Author Response

Pointwise response to reviewers’s comments on SUSTAINABILITY-1041237

Dear Sustainability Editor, Dear Reviewers,

Thank you for the reviewer comments for Suatainability-1041237; they are very helpful to improve our manuscript. Please find attached the revised article entitled “Melatonin as Master regulator in Plant Growth, Development and Stress Alleviator for Sustainable Agricultural Production: Current Status and Future Perspectives, for consideration for publication in Sustainability.

 Reviewer-2 comment is indicated below, together with the response (clarifications/ revisions) we have included in the revision. Revisions appear in track changings in the revised manuscript.

It is hoped that this revision satisfies the query raised by respected reviewer and manuscript will be considered for publication in Biomolecules.

With kind regards

Dr. Sumaira Anjum

Reviewer-2

Comments: The authors improved the manuscript well. I am satisfied with most of the authors' responses to comments, but I have still one remark:

In my opinion, there is a contradiction between the two paragraphs in Lines 674-684 and in Lines 685-697 regarding the health benefits of melatonin for human and animals. In the first paragraph the authors write that “As an antioxidant, the high levels of melatonin in plants are beneficial to its consumers including animals and human” and “may improve the human health”. And in second paragraph, the authors write that “implementation of melatonin in agriculture practice is still limited due to …. (1) melatonin is classified as a health hazard substance (code H-361) in terms of reproductive toxicity by EC (European Community) regulation…”. I suggest correcting the phrase “(1) melatonin is classified as a health hazard substance (code H-361) in terms of reproductive toxicity by EC (European Community) regulation” by adding information about the non-hazardous effect of melatonin with respect to human health, in general, and mutagenicity and carcinogenicity, in particular.

AUTHORS:  Dear Sir, thank you very much for giving your valuable suggestions, which helped us a lot to improve the quality of our manuscript. Sir we have corrected the sentence, “melatonin is classified as a health hazard substance (code H-361) in terms of reproductive toxicity by EC (European Community) regulation”, by adding the non-hazardous effects of melatonin with respect to human health in terms of the oral, dermal, inhalation and irritation categories, and in regards to mutagenicity and carcinogenicity in line 696-702 in revised manuscript.

This manuscript is a resubmission of an earlier submission. The following is a list of the peer review reports and author responses from that submission.